# Coumarins with Different Substituents from *Leonurus japonicus* Have Opposite Effects on Uterine Smooth Muscle

**DOI:** 10.3390/ijms251810162

**Published:** 2024-09-21

**Authors:** Yunqiu Fan, Chenhao Liu, Fang Wang, Lei Li, Yuqin Guo, Qinmei Zhou, Liang Xiong

**Affiliations:** 1State Key Laboratory of Southwestern Chinese Medicine Resources, School of Pharmacy, Chengdu University of Traditional Chinese Medicine, Chengdu 611137, China; fanyunqiuch@163.com (Y.F.); lileilei9923@163.com (L.L.); gyqthunder@163.com (Y.G.); 2School of Basic Medical Sciences, Chengdu University of Traditional Chinese Medicine, Chengdu 611137, China; liuch930512@163.com; 3Shandong Academy of Pharmaceutical Sciences, Jinan 250101, China; wangfang-zy@sdaps.cn; 4Institute of Traditional Chinese Medicine Innovation, Chengdu University of Traditional Chinese Medicine, Chengdu 611137, China

**Keywords:** *Leonurus japonicus*, coumarins, uterine smooth muscle contraction, promoting effect, inhibiting effect, preliminary mechanism

## Abstract

*Leonurus japonicus* Houtt is an exceptional medicinal herb used to treat obstetrical and gynecological diseases in traditional Chinese medicine, and it has significant effects on the treatment of dysmenorrhea and postpartum hemorrhage. This study investigated the effects of coumarins with diverse substituent groups from *L. japonicus* on isolated uterine smooth muscle and the preliminary mechanism of the most effective compound. Eight coumarins isolated from *L. japonicus* were assessed for their effects on the isolated uterine smooth muscle of nonpregnant rats in vitro. Coumarins **1** and **2** significantly promoted the contraction of rat uterine smooth muscle strips, whereas coumarins **3**–**5** showed remarkable relaxing effects against oxytocin (OT)-induced rat uterine smooth muscle contraction. Further mechanism investigations revealed that bergapten (coumarin **1**) significantly increased the level of Ca^2+^ in uterine tissues by promoting extracellular Ca^2+^ influx and intracellular Ca^2+^ release, which were related to the activation of L-type Ca^2+^ channels and α-receptors. By contrast, osthole (coumarin **5**), an α receptor antagonist, inhibited OT-induced uterine smooth muscle contraction by decreasing the level of Ca^2+^ in uterine tissues via inhibition of extracellular Ca^2+^ influx and intracellular Ca^2+^ release. This study demonstrates that the coumarins from *L. japonicus* are effective substances for regulating uterine smooth muscle contraction, but different coumarins with diverse substituent groups have different, even opposite effects. It can be inferred that coumarins are closely related to the efficacy of *L. japonicus* in the treatment of dysmenorrhea and postpartum hemorrhage.

## 1. Introduction

*Leonurus japonicus* Houtt (Chinese motherwort) is an exceptional medicinal herb used to treat obstetrical and gynecological diseases in traditional Chinese medicine (TCM) [1]. Extensive research has shown that more than 300 TCM prescriptions containing *L. japonicus* have been used for regulating menstrual disturbance, dysmenorrhea, amenorrhea, blood stasis, and postpartum hemorrhage [2,3,4,5]. Modern research has shown that motherwort has many pharmacological effects, such as the regulation of uterine contraction [6], endometrial reparation [1], pro-angiogenesis [7], anti-platelet aggregation [8], cardiovascular protection [9], anti-inflammation [10], and neuroprotection [11]. Interestingly, motherwort has bidirectional regulatory effects on uterine contraction activity [12], which are closely related to its efficacy in treating dysmenorrhea and postpartum hemorrhage [13]. Although efforts have been made to elucidate the effective constituents of motherwort, there is a limited understanding of the substances that regulate uterine activity. In our previous study, cyclopeptides and alkaloids isolated from motherwort were shown to promote the contraction of isolated uterine smooth muscle from rats, whereas flavonoid glycosides exhibited the opposite effect [6]. To date, the effects of other types of motherwort compounds on uterine smooth muscle have not been evaluated.

Coumarins are derivatives of cinnamic acid with a benzo-α-pyrone skeleton, which are widespread in plants and have a variety of biological activities, such as anti-platelet aggregative, antibacterial, and vasodilator effects [14,15]. An increasing number of studies have found that several coumarins can affect smooth muscle contractility in vitro [16,17,18]. Although a series of coumarins have been isolated and identified from motherwort [1,8], their effects on uterine smooth muscle have not been explored [19,20,21]. This study determined whether coumarins isolated from motherwort (Table 1) have an effect on uterine smooth muscle in vitro. In addition, the preliminary mechanisms of effective coumarins were investigated.

## 2. Results

### 2.1. Effects of Coumarins on OT-Induced Rat Uterine Smooth Muscle Contraction

OT (Oxytocin), a uterine smooth muscle stimulant, can induce strong and sustained contraction of uterus, which is similar to the intense contraction of dysmenorrhea. Thus, it is often used as a model drug in anti-dysmenorrhea experiments [22]. As shown in (Figure 1), coumarins **3**–**5** at 50 μM significantly inhibited the OT-induced contraction of rat uterine smooth muscle strips, with maximum inhibition rates of 16.63 ± 4.05% (*p* < 0.05), 38.95 ± 4.84% (*p* < 0.01), and 61.49 ± 5.27% (*p* < 0.01), respectively. However, coumarins **1** and **2** markedly stimulated the uterine smooth muscle contraction, and their inhibition rates were calculated as −40.13 ± 3.47% (*p* < 0.01) and −37.13 ± 8.03% (*p* < 0.01), respectively. These results indicate that coumarins with different substituents from *L. japonicus* have opposite effects on uterine smooth muscle.

### 2.2. Excitatory Effect of Coumarin 1 on Rat Uterine Smooth Muscle Contraction and Its Mechanism

#### 2.2.1. Excitatory Effect of Coumarin 1 on Normal Rat Uterine Smooth Muscle Strips

Normal rat uterine smooth muscle strips were used to further confirm the excitability of coumarin **1** on uterus. As shown in Figure 2, coumarin **1** at concentrations of 1, 10, and 50 μM significantly promoted the uterine smooth muscle contraction. The average promotion rates of contractile tension were 10.34 ± 2.85% (*p* > 0.05), 27.89 ± 8.36% (*p* < 0.05), and 51.04 ± 10.10% (*p* < 0.01), and the average promotion rates of contractile activity were 10.34 ± 2.85% (*p* > 0.05), 56.65 ± 21.92% (*p* < 0.01), and 162.32 ± 45.51% (*p* < 0.01).

#### 2.2.2. Effect of Coumarin 1 on Rat Uterine Smooth Muscle Strips Pretreated with Inhibitors

The functional status of receptors and ion channels in uterine smooth muscle can affect the contractile activity of uterus. In this study, we used an L-type Ca^2+^ channel blocker (VER, verapamil hydrochloride), a prostaglandin synthase inhibitor (IND, indomethacin), an H_1_ receptor blocker (DIP, diphenhydramine), and an α receptor blocker (PHE, phentolamine) to explore the possible mechanisms by which coumarin **1** promotes uterine smooth muscle contraction. As shown in Figure 3, the excitatory effect of coumarin **1** on uterine smooth muscle was significantly suppressed by VER and PHE (*p* < 0.01), whereas IND and DIP could not affect the effect of coumarin **1**. These results suggested that coumarin **1** worked through the L-type Ca^2+^ channel and α receptor but not through the H_1_ receptor or synthesis/release of prostaglandin.

#### 2.2.3. Effect of Coumarin 1 on Extracellular Ca^2+^ Influx and Intracellular Ca^2+^ Release

To further explore the effect of coumarin **1** on internal Ca^2+^ release and external Ca^2+^ influx, we performed the experiments with Ca^2+^-free and recalcification solutions. First, in a Ca^2+^-free solution, coumarin **1** has a significant excitatory effect on rat uterine smooth muscle (Figure 4). At 10 and 50 μM, the promotion rates of contractile tension were 59.34 ± 8.71% (*p* < 0.01 vs. control) and 106.78 ± 14.56% (*p* < 0.01 vs. control), respectively. This excitatory effect can be considered to be caused by the promotion of intracellular Ca^2+^ release. Then, after recalcification of the solution, the excitatory effect of coumarin **1** was further improved at 10 and 50 μM (36.46 ± 2.73% [*p* < 0.01 vs. control] and 66.45 ± 6.69% [*p* < 0.01 vs. control], respectively), which indicated that coumarin **1** also promoted extracellular Ca^2+^ influx in uterine smooth muscle cells.

### 2.3. Inhibitory Effect of Coumarin 5 on Rat Uterine Smooth Muscle Contraction and Its Mechanism

#### 2.3.1. Inhibitory Effect of Coumarin 5 on OT-Induced Rat Uterine Smooth Muscle Contraction

To explore the inhibitory activity of coumarin **5** on uterus contraction, OT was used to induce the severe contraction of rat uterine smooth muscle strips. As shown in Figure 5, coumarin **5** at 10 and 50 μM significantly inhibited the OT-induced contraction of uterine smooth muscle, with inhibition rates of 33.05 ± 2.22% (*p* < 0.01 vs. control) and 61.49 ± 5.27% (*p* < 0.01 vs. control), respectively.

#### 2.3.2. Effect of Coumarin 5 on Rat Uterine Smooth Muscle Strips Pretreated with Inhibitors

The possible mechanism of coumarin **5** was preliminarily investigated using the same inhibitors (VER, IND, DIP, and PHE) as described for coumarin **1**. The results indicated that pretreatment with PHE significantly decreased the relaxant effect of coumarin **5** on uterine smooth muscle, whereas other inhibitors did not change the effect of coumarin **5** (Figure 6). Thus, the inhibitory effect of coumarin **5** on uterus contraction involved the α receptor.

#### 2.3.3. Effect of Coumarin 5 on Extracellular Ca^2+^ Influx and Intracellular Ca^2+^ Release

##### Inhibitory Effect of Coumarin 5 on Uterine Smooth Muscle Contraction Induced by K^+^ Depolarization

In this study, KCl was used to create a high K^+^ condition for depolarization of the membrane potential, which could open the voltage-dependent Ca^2+^ channel, causing extracellular Ca^2+^ influx. As shown in Figure 7, coumarin **5** (10 and 50 μM) showed significant inhibitory effects on rat uterine smooth muscle contraction induced by KCl. These results suggest that coumarin **5** can target the voltage-gated calcium channels to inhibit extracellular Ca^2+^ influx and decrease the [Ca^2+^]_i_.

##### Inhibitory Effect of Coumarin 5 on Uterine Smooth Muscle Contraction Induced by OT in a Ca^2+^-Free Solution

First, the rat uterine smooth muscle strips showed no obvious spontaneous contraction in a Ca^2+^-free solution. Then, OT (0.008 U/mL) was added to the solution to increase the uterine smooth muscle contraction (Figure 8). After treatment with coumarin **5** at 1, 10, and 50 μM, the contractile tension was markedly inhibited, with inhibition rates of 45.45 ± 3.40% (*p* < 0.01 vs. control), 56.93 ± 4.10% (*p* < 0.01 vs. control), and 74.84 ± 7.65% (*p* < 0.01 vs. control), respectively. These results indicated that coumarin **5** could inhibit the Ca^2+^ release from the sarcoplasmic reticulum to decrease the [Ca^2+^]_i_.

## 3. Discussion

Motherwort has been used as an important medicine for treating obstetrical and gynecological diseases in China for more than 1,800 years; it has specifically been used to alleviate postpartum hemorrhage and dysmenorrhea [1]. According to the modern medical theory, treatment of postpartum hemorrhage is mainly related to the promotion of uterine contraction [23], whereas dysmenorrhea can be treated by relaxing uterine smooth muscle [24]. In this study, motherwort coumarins were systemically studied to understand how they regulate uterine contraction. The findings revealed that some coumarins exhibited significant effects on regulating uterine smooth muscle strips. Notably, motherwort coumarins with different substituents showed opposite effects on uterine smooth muscle. Specifically, coumarins **1** and **2** promoted uterine contraction, whereas coumarins **3**–**5** had inhibitory effects. Comparisons of the structures and effects of coumarins **1**−**4** indicated that the contractile activity of furocoumarins in motherwort is closely related to the substitution of OCH_3_-5. This is an unusual phenomenon in nature, i.e., for compounds with similar structures to have completely opposite activities [25]. For example, 20(*S*)-protopanaxadiols have significant antihemolytic effects [26], whereas 20(*S*)-protopanaxatriols induce hemolysis [27]. Notably, these interesting opposite effects on uterine contraction of motherwort coumarins are associated with the clinical use of motherwort for the treatment of postpartum hemorrhage and dysmenorrhea, which enrich the understanding of active substances in motherwort.

Coumarins **1** and **2** not only promoted the contraction of normal uterine smooth muscle strips but also enhanced the OT-induced uterine smooth muscle contraction, which suggested a synergistic effect of these coumarins with OT. Although OT is a uterotonic drug known for treating postpartum hemorrhage, its receptor saturation and weak contraction effect on the lower segment of uterus limit its clinical efficacy [28] Thus, many studies have investigated the clinical efficacy of a combination of OT and other drugs, such as motherwort preparations. It is well known that uterine contraction is closely related to the concentration of Ca^2+^ in uterine smooth muscle cells [29]. The contraction of uterine smooth muscle is facilitated by pivotal mechanisms involving L-type calcium channels, α-receptors, histamine H_1_ receptors, and the synthesis and exocytosis of prostaglandins [24,30,31,32]. To further investigate the mechanism of coumarin **1**, four inhibitors were used in this study. VER is an inhibitor of L-type Ca^2+^ channels, IND is an inhibitor of prostaglandin synthase, DIP is a blocker of H_1_ receptor, and PHE is a nonselective α-receptor blocker. Among them, only VER and PHE inhibited the excitatory effect of coumarin **1** on uterine smooth muscle, which suggested that coumarin **1** could activate L-type Ca^2+^ channels and α-receptors to promote uterine contraction. Further studies confirmed that coumarin **1** could significantly increase the level of Ca^2+^ in uterine tissues by promoting extracellular Ca^2+^ influx and intracellular Ca^2+^ release.

Many studies have found that several medicines can treat dysmenorrhea by relaxing the uterine smooth muscle. In this study, three coumarins (**3**−**5**) exhibited remarkable relaxant activity on uterine smooth muscle, which may be related to the anti-dysmenorrhea property of motherwort. Notably, although coumarins **4**−**8** all contain isopentenyl or isopentyl group at C-8, only coumarins **4** and **5** were active. Substitution and oxidation may be the main reasons for this difference in relaxant activity of coumarins on uterine smooth muscle. In addition, the four above-mentioned inhibitors were also used to further study the inhibitory effect of coumarin **5** on uterine contraction. The results showed that only PHE significantly decreased the effect of coumarin **5**. Thus, coumarin **5** could play a role in relaxing the uterine smooth muscle via the α receptor, rather than the H_1_ receptor, L-type Ca^2+^ channel, or prostaglandin release. Moreover, this study confirmed that coumarin **5** could reduce the level of Ca^2+^ in uterine tissues by inhibiting extracellular Ca^2+^ influx and intracellular Ca^2+^ release. Interestingly, the opposite activities of coumarins **1** and **5** are both related to the α receptor. It is known that two primary subtypes of α-receptor, α_1_ and α_2_, play distinct roles in uterine smooth muscle. Activation of the α_1_ receptor often triggers muscle contractions, while the α_2_ receptor may prevent muscle contractions or modulate their response, exhibiting a more complex regulatory function. Therefore, it is plausible that the mechanisms of coumarins **1** and **5** might be attributed to their interaction with distinct subtypes of α receptor or the intrinsic activity of the receptor targeted by the two coumarins. Taken together, these findings provide evidence for opposite effects and underlying mechanisms of motherwort coumarins on uterine contraction.

## 4. Materials and Methods

### 4.1. Animals

Nonpregnant female Sprague-Dawley rats, weighing 180–220 g, were provided by DaShuo Biotechnology Co., Ltd. (Chengdu, China). All rats were reared in a 12:12 h light/dark cycle environment with a relative humidity of 50 ± 5% and temperature of 25 ± 1 °C. Food and water were administered ad libitum during the study period. All experiments were approved by the Experimental Animal Ethics Committee of Chengdu University of Traditional Chinese Medicine (Approval No. 2020–04).

### 4.2. Chemicals

Estradiol valerate tablets were purchased from Delpharm Lille S.A.S. (Chengdu, China). NaCl, NaHCO_3_, KCl, and other chemicals for Locke’s solution were obtained from Chengdu Kelong Chemical Co., Ltd. (Chengdu, China). VER, IND, DIP, and PHE were purchased from Selleck (Shanghai, China).

### 4.3. Preparation of Uterine Smooth Muscle Strips

Female unfertilized healthy adult Sprague-Dawley rats were selected for experiments and were intragastrically administered estradiol valerate tablets (8 mg/kg) for 2 consecutive days. After the rats were sacrificed by cervical dislocation, the uterine tissue was isolated quickly and placed in a container containing Locke’s solution (157 mM NaCl, 5.6 mM KCl, 2.2 mM CaCl_2_, 1.8 mM NaHCO_3_, and 5.6 mM glucose). Adherent fat and connective tissue were carefully removed from the uterus. Then, the uterine muscle was cut into strips of the same length and placed in an isolated organ bath containing Locke’s solution and bubbled with 95% O_2_ and 5% CO_2_ at 37 °C. The uterine muscle strips were preloaded with 1 g tension and equilibrated for at least 30 min to obtain a stable spontaneous contraction. The uterine contraction curves were recorded by a 16-channel physiological recording system (ADInstruments Shanghai Trading Co., Ltd., Shanghai, China) [33,34,35].

### 4.4. Effects of Coumarins on the Uterine Smooth Muscle Strips

After equilibration in Locke’s solution for 30 min, the uterine muscle strips were induced to contract with OT for 10 min. Subsequently, each coumarin (**1**–**8**) dissolved in DMSO [32] was added to the bath and monitored for another 10 min. The contractile tension and frequency were recorded, and the inhibition rates of contractile tension and activity were calculated as follows: inhibition rate of contractile tension = (model mean of contractile tension – test mean of contractile tension)/model mean of contractile tension × 100%; inhibition rate of contractile activity = (model mean of contractile tension × frequency) – (test mean of contractile tension × frequency)/(model mean of contractile tension × frequency) × 100%. DMSO (0.2%) served as the negative control, and VER was used as the positive control [31]. In addition, coumarin **1** with the greatest excitatory effect on OT-induced uterine smooth muscle contraction was further evaluated for its effect on normal uterine smooth muscle.

### 4.5. Effects of Coumarins 1 and 5 on Uterine Smooth Muscle Strips Pretreated with Inhibitors

The normal uterine smooth muscle strips were equilibrated in Locke’s solution for 30 min. Next, the strips were administered by inhibitors (1 × 10^−7^ M VER, 1 × 10^−5^ M IND, 1 × 10^−6^ M DIP, or 1 × 10^−6^ M PHE) for 10 min. Then, coumarin **1** with an excitatory effect on uterine smooth muscle contraction was added to the bath and monitored for 10 min [30], and 0.2% DMSO served as the negative control. The contractile tension was recorded.

Regarding coumarin **5**, which had an inhibitory effect on uterine smooth muscle contraction, the above-mentioned inhibitors were used to pretreat the uterine muscle strips for 10 min. Then, OT was added to induce uterine smooth muscle contraction for another 10 min. Subsequently, coumarin **5** was added to the bath, and the uterine smooth muscle strips were monitored for 10 min. The following experimental procedures and data records were performed as previously described.

### 4.6. Effects of Coumarins 1 and 5 on External Calcium Influx and Internal Calcium Release in Uterine Smooth Muscle

#### 4.6.1. Excitatory Effect of Coumarin 1 on Uterine Smooth Muscle Contraction in a Ca^2+^-Free and Recal-Cification Solution

Briefly, the rat uterine smooth muscle strips were equilibrated in Locke’s solution for 30 min. Then, to assess the excitatory effect of coumarin **1** on uterine smooth muscle contraction by promoting intracellular calcium release, the uterine smooth muscle strips were preincubated in Ca^2+^-free solution containing EGTA for 10 min. Then, coumarin **1** was added to this solution, and the contractile tension was recorded for 10 min. Subsequently, to assess the excitatory effect of coumarin **1** on uterine smooth muscle contraction via extracellular calcium influx, Ca^2+^ was added to the solution at a final concentration of 2.2 mM. After 10 min of recalcification, the effect of coumarin **1** was tested [31,34]. DMSO (0.2%) served as the negative control.

#### 4.6.2. Inhibitory Effect of Coumarin 5 on Uterine Smooth Muscle Contraction Induced by K^+^ Depolarization

The uterine muscle strips were equilibrated in Locke’s solution for 30 min. Then, KCl was added to induce a rapid contraction of the uterine muscle strips at a final concentration of 60 mM, followed by slight relaxation and a plateau period of contraction. Subsequently, the inhibitory effect of coumarin **5** at different concentrations on uterine smooth muscle contraction was monitored for 10 min [36]. The contractile tension was recorded, and the inhibition rate of contractile tension was calculated.

#### 4.6.3. Inhibitory Effect of Coumarin 5 on OT-Induced Uterine Smooth Muscle Contraction in Ca^2+^-Free Solution

After the rat uterine smooth muscle strips were equilibrated in Locke’s solution for 30 min, the solution was replaced by a Ca^2+^-free solution containing 0.3 mM EGTA. After equilibration, OT was added to make the strips shrink continuously for 10 min. Then, coumarin **5** was added to the bath and monitored for 10 min [36]. The contractile tension was recorded, and the inhibition rate of contractile tension was calculated.

### 4.7. Statistical Analysis

All results are expressed as the mean ± standard deviation (SD). All experiments were conducted at least six times. Figures were prepared using GraphPad Prism software version 5.0 (GraphPad Software, Inc., San Diego, CA, USA). The paired sample *t*-test was used to compare the differences, and *p* < 0.05 was considered statistically significant.

## 5. Conclusions

Motherwort is an excellent TCM for treating postpartum hemorrhage and dysmenorrhea due to its regulatory effect on uterine smooth muscle contraction. This is the first study to systemically investigate the effects of motherwort coumarins on uterine contraction. Among the eight tested coumarins, coumarins **1** and **2** exhibited remarkable excitatory effects on rat uterine smooth muscle strips, whereas coumarins **3**–**5** had the opposite effects. Coumarins **1** and **5** were found to be the most active compounds. Further studies using inhibitors indicated that coumarin **1** could activate the L-type Ca^2+^ channel and α-receptor, whereas coumarin **5** was an α receptor antagonist. In addition, this study confirmed that coumarin **1** significantly promoted extracellular Ca^2+^ influx and intracellular Ca^2+^ release to increase [Ca^2+^]_i_, whereas coumarin **5** had opposite effects on Ca^2+^ influx and release. These interesting results reveal that motherwort coumarins with different substituents have opposite effects on uterine smooth muscle and greatly contribute to the regulation of uterine smooth muscle.

## Figures and Tables

**Figure 1 ijms-25-10162-f001:**
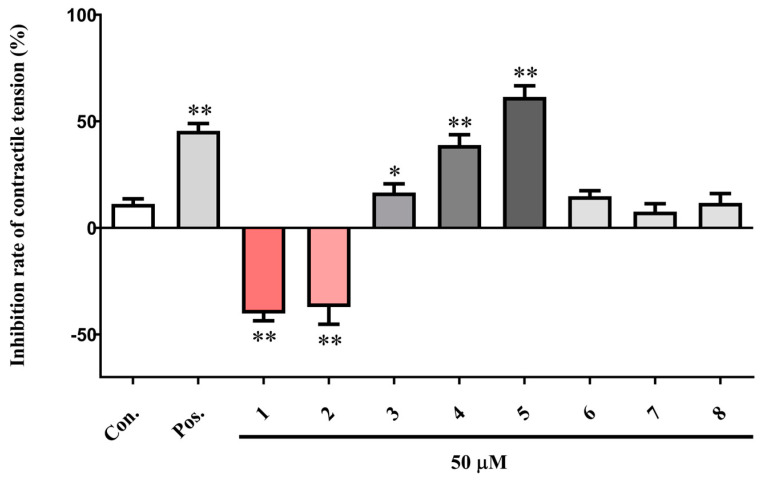
Effects of eight coumarins (**1**–**8**) on oxytocin-induced rat uterine smooth muscle contraction. The uterine contraction tensions were recorded by a physiological recording system. The data are expressed as the mean ± SD (n = 6). * *p* < 0.05 and ** *p* < 0.01 vs. control group. The negative control group was treated with 0.2% DMSO, and the positive control group was treated with 0.5 μM verapamil hydrochloride. Con.: negative control group, Pos.: positive control group.

**Figure 2 ijms-25-10162-f002:**
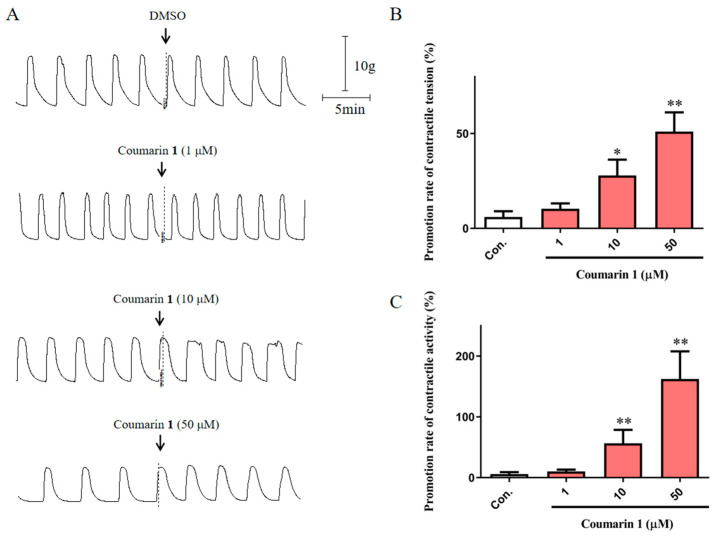
Excitatory effects of coumarin **1** (1, 10, and 50 μM) on normal rat uterine smooth muscle strips. (**A**) Traces for the effects of DMSO and coumarin **1** on rat uterine smooth muscle contraction; (**B**) Promotion rates of contractile tension of DMSO and coumarin **1**; (**C**) Promotion rates of contractile activity of DMSO and coumarin **1**. The negative control group was treated with 0.2% DMSO. The data are expressed as the mean ± SD (n = 6). * *p* < 0.05 and ** *p* < 0.01 vs. control group. Con.: negative control group.

**Figure 3 ijms-25-10162-f003:**
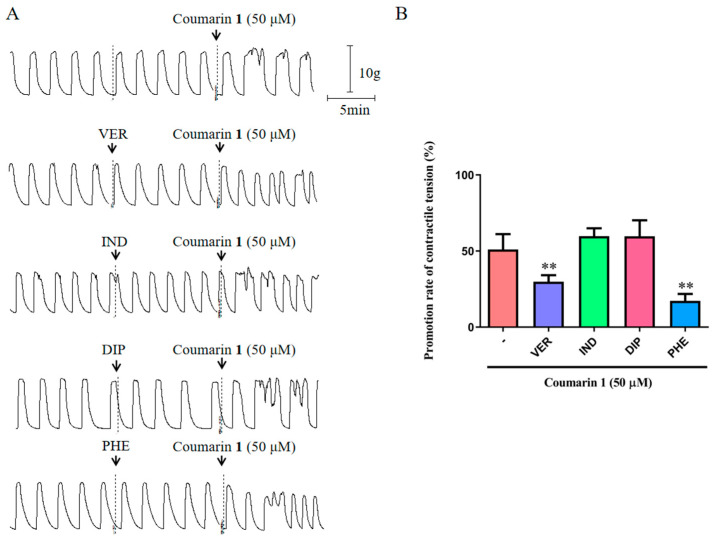
Excitatory effect of coumarin **1** (50 μM) on rat uterine smooth muscle strips pretreated with inhibitors. (**A**) Traces for the contraction of rat uterine smooth muscle strips treated with coumarin **1** with or without inhibitors (1 × 10^−7^ M VER, 1 × 10^−5^ M IND, 1 × 10^−6^ M DIP, or 1 × 10^−6^ M PHE); (**B**) Promotion rates of contractile tension of coumarin **1** with or without inhibitors. The data are expressed as the mean ± SD (n = 6). ** *p* < 0.01 vs. coumarin **1** group.

**Figure 4 ijms-25-10162-f004:**
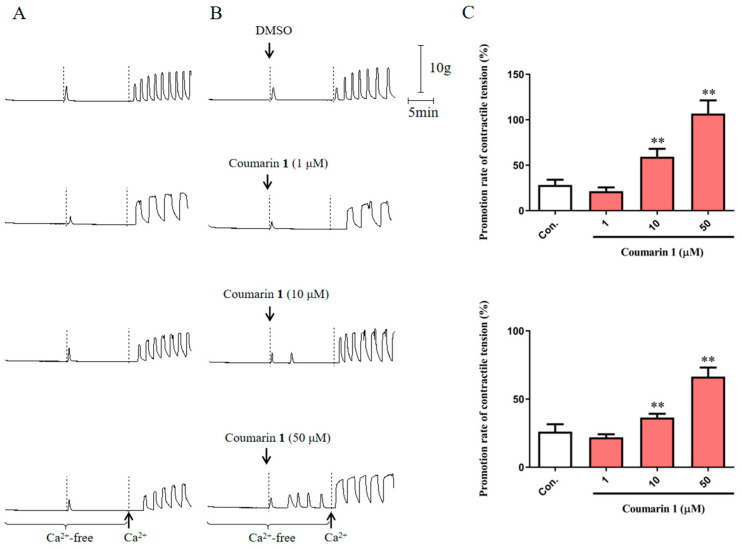
Excitatory effect of coumarin **1** (1, 10, and 50 μM) on rat uterine smooth muscle in Ca^2+^-free and recalcification solutions. (**A**) Traces for the contraction of rat uterine smooth muscle strips without drugs; (**B**) Traces for the contraction of rat uterine smooth muscle strips treated with DMSO or coumarin **1**; (**C**) Promotion rates of contractile tension of DMSO and coumarin **1**. The negative control group was treated with 0.2% DMSO. The data are expressed as the mean ± SD (n = 6). ** *p* < 0.01 vs. control group. Con.: negative control group.

**Figure 5 ijms-25-10162-f005:**
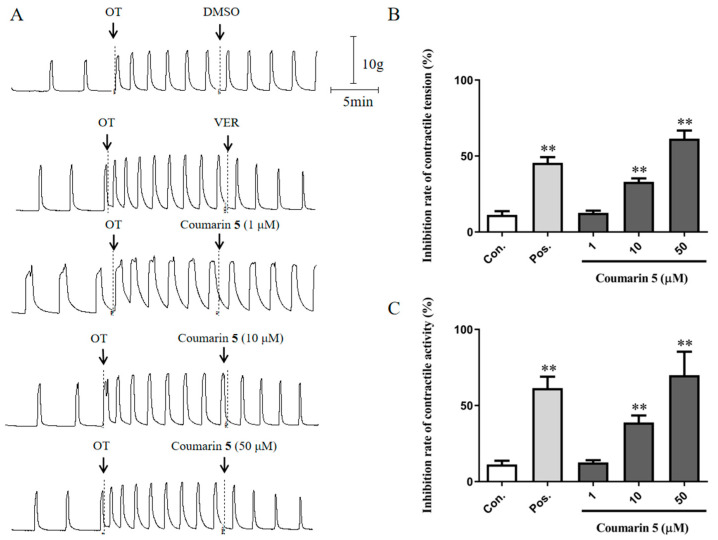
Inhibitory effect of coumarin **5** (1, 10, and 50 μM) on OT-induced rat uterine smooth muscle contraction. (**A**) Traces for the contraction of rat uterine smooth muscle strips treated with DMSO, VER, or coumarin **5**; (**B**) Inhibition rates of contractile tension of DMSO, VER, and coumarin **5**; (**C**) Inhibition rates of contractile activity of DMSO, VER, and coumarin **5**. The negative control group was treated with 0.2% DMSO, and the positive control group was treated with 0.5 μM VER. The data are expressed as the mean ± SD (n = 6). ** *p* < 0.01 vs. control group. Con.: negative control group, Pos.: positive control group.

**Figure 6 ijms-25-10162-f006:**
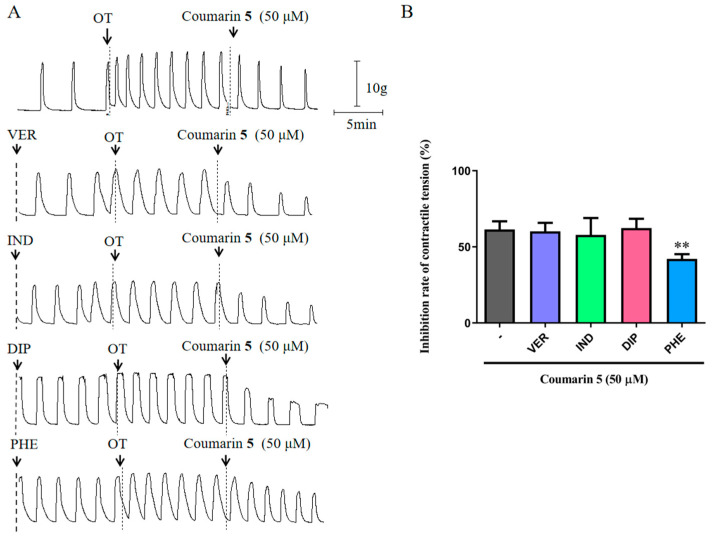
Inhibitory effect of coumarin **5** (50 μM) on rat uterine smooth muscle strips pretreated with inhibitors (1 × 10^−7^ M VER, 1 × 10^−5^ M IND, 1 × 10^−6^ M DIP, or 1 × 10^−6^ M PHE). (**A**) Traces for the contraction of rat uterine smooth muscle strips treated with coumarin **5** with or without inhibitors; (**B**) Inhibition rates of contractile tension of coumarin **5** with or without inhibitors. The data are expressed as the mean ± SD (n = 6). ** *p* < 0.01 vs. coumarin **5** group.

**Figure 7 ijms-25-10162-f007:**
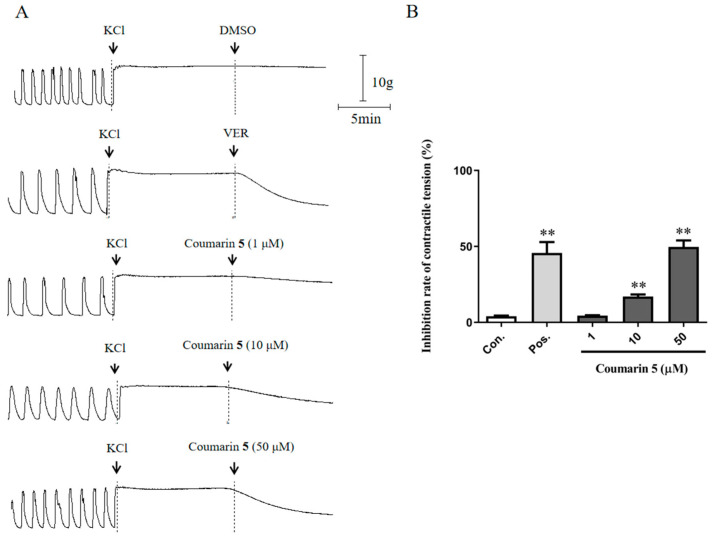
Inhibitory effect of coumarin **5** (1, 10, and 50 μM) on KCl-induced uterine smooth muscle contraction. (**A**) Traces for the KCl-induced contraction of rat uterine smooth muscle strips treated with DMSO, VER, or coumarin **5**; (**B**) Inhibition rates of contractile tension of DMSO, VER, and coumarin **5**. The negative control group was treated with 0.2% DMSO, and the positive control group was treated with 0.5 μM VER. The data are expressed as the mean ± SD (n = 6). ** *p* < 0.01 vs. control group. Con.: negative control group, Pos.: positive control group.

**Figure 8 ijms-25-10162-f008:**
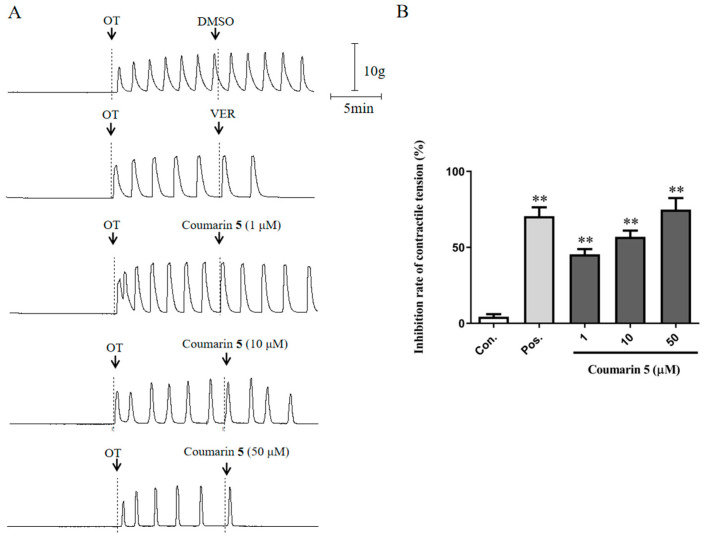
Inhibitory effect of coumarin **5** (1, 10, and 50 μM) on OT-induced uterine smooth muscle contraction in a Ca^2+^-free solution. (**A**) Traces for the OT-induced contraction of rat uterine smooth muscle strips treated with DMSO, VER, or coumarin **5**; (**B**) Inhibition rates of contractile tension of DMSO, VER, and coumarin **5**. The negative control group was treated with 0.2% DMSO, and the positive control group was treated with 0.5 μM VER. The data are expressed as the mean ± SD (n = 6). ** *p* < 0.01 vs. control group. Con.: negative control group, Pos.: positive control group.

**Table 1 ijms-25-10162-t001:** Coumarins Isolated From Motherwort.

NO.	Name	Molecular Formula	CAS	Structure
**1**	Bergapten	C_12_H_8_O_4_	484-20-8	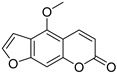
**2**	Isopimpinellin	C_13_H_10_O_5_	482-27-9	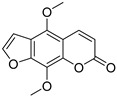
**3**	8-Methoxypsoralen	C_12_H_8_O_4_	298-81-7	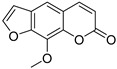
**4**	Imperatorin	C_16_H_14_O_4_	482-44-0	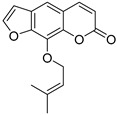
**5**	Osthole	C_15_H_16_O_3_	484-12-8	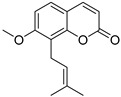
**6**	Murrayone	C_15_H_14_O_4_	19668-69-0	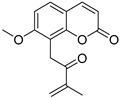
**7**	Isomerazin	C_15_H_16_O_4_	1088-17-1	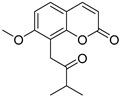
**8**	Meranzin hydrate	C_15_H_18_O_5_	5875-49-0	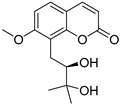

## Data Availability

Data are contained within the article.

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
