# Peer review of "Coumarins with Different Substituents from *Leonurus japonicus* Have Opposite Effects on Uterine Smooth Muscle"

_ijms, 2024, doi:10.3390/ijms251810162_

Round 1
Reviewer 1 Report
Comments and Suggestions for Authors
The manuscript submitted by Yun-Qiu Fan et al. presents a study investigating the effects of coumarins derived from Leonurus japonicus Houtt. (Chinese motherwort) on uterine smooth muscle (USM) contractions. The authors hypothesize that these effects may explain the traditional use of this plant in treating dysmenorrhea and postpartum hemorrhage. The methodology involves measuring changes in contraction tension and frequency induced by eight coumarins. Subsequently, the authors delved deeper into the mechanisms of the most potent contractile (number 1) and relaxant (number 5) coumarins using various pharmacological blockers. The aim of the study is interesting, but the data presentation is unclear and the statistical analysis is unproper.
Major Comments:
1- The meaning of 'mean of contractile tension' is unclear. This reviewer interprets it as the mean of the maximal force amplitude generated during transient contractions under each condition. However, in the representative recording in figure 2, the addition of coumarin 1 did not affect the amplitude of contractions. This finding is inconsistent with the mean data presented in panel B.
2- The authors claim a concentration-dependent effect for both coumarins in Figures 2, 4, 5, 7, and 8 based solely on comparisons to the vehicle control. To establish concentration-dependence more rigorously, the effects of different concentrations should be directly compared, or a complete concentration-response curve with Hill equation analysis should be provided.
3- Figures 4 and 8 demonstrate that uterine contractions depends on extracellular Ca2+ influx, as activity is negligible in a Ca2+-free solution. Conversely, Figure 3 shows that the addition of VER, a Ca2+ channel blocker, does not affect contractions, suggesting a different mechanism.
4- This reviewer suggests that the authors should investigate a potential competitive interaction between coumarin 1 and PHE to confirm its activity as an alpha-agonist.
5- The reduction of the relaxant effect of coumarin 5 by PHE is only compatible with an inverse agonist activity. Do the authors know if alpha receptors have intrinsic activity in this tissue?
6- Page 8: what is the meaning of “stabilizing membrane potential? Is really difficult to change membrane potential after the addition of 60 mM KCl.
Minor Comments:
1- The figure legends are insufficient. Essential details are missing, making it necessary to consult the methodology section for proper interpretation.
2- Provide a rationale for the selection of VER, DIP, PHE, and IND for the mechanistic studies.
Author Response
Comment 1: The meaning of 'mean of contractile tension' is unclear. This reviewer interprets it as the mean of the maximal force amplitude generated during transient contractions under each condition. However, in the representative recording in figure 2, the addition of coumarin 1 did not affect the amplitude of contractions. This finding is inconsistent with the mean data presented in panel B.
Response 1: Thank you for your comment. “The mean of contractile tension” is not simply “the mean of the maximal force amplitude” generated during transient contractions under each condition. Instead, it is measured by considering both the amplitude and the frequency of these contractions together. In this experiment, the instrument recorded the contraction curve of uterine smooth muscle strips and directly read the mean contractile amplitude (g), contractile frequency (n), duration of contractile waves (t/s), and mean contractile tension (g.s). The statistical data presented in this manuscript is the average contractile tension. Although the fluctuations in the contractile amplitude were not pronounced, significant changes in the contractile frequency and width of the contraction peaks could be observed. Thus, in Figure 2, the findings from the representative curves are in agreement with the data displayed in panel B, despite the fact that the addition of coumarin 1 did not significantly affect the contractile amplitude.
Comment 2: The authors claim a concentration-dependent effect for both coumarins in Figures 2, 4, 5, 7, and 8 based solely on comparisons to the vehicle control. To establish concentration-dependence more rigorously, the effects of different concentrations should be directly compared, or a complete concentration-response curve with Hill equation analysis should be provided.
Response 2: Special thanks for your good suggestion. We investigated the effects of coumarin 1 and 5 at three concentrations of 1, 10, and 50 μM. The contractile tension was inhibited in a dose-dependent manner based on the inhibition rates. Then, we mainly explored the underlying mechanisms. Thus, the concentration for the best effect was chosen for further studies as shown in Figures 3 and 6. Indeed, a complete concentration-response curve would more rigorously confirm the concentration dependency. Thank you very much for helping us to improve our results.
Comment 3: Figures 4 and 8 demonstrate that uterine contractions depends on extracellular Ca2+ influx, as activity is negligible in a Ca2+-free solution. Conversely, Figure 3 shows that the addition of VER, a Ca2+ channel blocker, does not affect contractions, suggesting a different mechanism.
Response 3: Thank you for this comment. As shown in Figure 3, VER (a calcium channel blocker) at a concentration of 1×10-7 M did not affect the excitatory effect of coumarin 1 on rat uterine smooth muscle strips. VER at 1×10-7 M was commonly used for investigating whether the effects of compounds on uterine contractions are related to calcium channels [1,2]. However, in Figure 4, a solution devoid of calcium was employed to delve further into whether the effect of coumarin 1 on uterine contraction was contingent upon alterations in intracellular calcium ion concentrations. In Figure 8, the inhibitory effect of coumarin 5 on oxytocin-induced uterine smooth muscle contraction was assessed, and VER (0.5 μM) was used as a positive control. These experiments are fundamentally different, each employing different models.
Reference
[1] Liu, H., Zhu, T., Ma, Y., Qu, S., 2003, Effect of erythromycin on contractile response of uterine smooth muscle strips in non-pregnant rats, Pol. J. Pharmacol. 55, 57–62.
[2] Xu, J., Wang, W., Qin, X., 2005, Different mechanisms mediated the potentiation uterine smooth muscle caused by Polygala tenuifolia decoction progesterone in virginal rats, Chin. J. Clin. Mater. Med. 30, 295–297.
Comment 4: This reviewer suggests that the authors should investigate a potential competitive interaction between coumarin 1 and PHE to confirm its activity as an alpha-agonist.
Response 4: Thank you very much for the good suggestion. In this study, we used an L-type Ca2+ channel blocker (VER, verapamil), a prostaglandin synthase inhibitor (IND, indomethacin), an H1 receptor blocker (DIP, diphenhydramine), and an α receptor blocker (PHE, phentolamine) to explore the possible mechanisms by which coumarin 1 promotes uterine smooth muscle contraction. As shown in Figure 3, the excitatory effect of coumarin 1 on uterine smooth muscle was significantly suppressed by VER and PHE (P < 0.01), whereas IND and DIP could not affect the effect of coumarin 1. These results suggested that coumarin 1 could activate L-type Ca2+ channels and α-receptors to promote uterine contraction. We are going to carry out in-depth exploration and target validation in further studies. Thanks again for the meaningful comment.
Comment 5: The reduction of the relaxant effect of coumarin 5 by PHE is only compatible with an inverse agonist activity. Do the authors know if alpha receptors have intrinsic activity in this tissue?
Response 5: Thanks for your good suggestion. Modern studies provide scientific evidence regarding the presence and functions of α-receptors in uterine smooth muscle, serving as important references for understanding the roles of these receptors in reproductive physiology and pathophysiology [3-5]. Our results showed that only PHE significantly decreased the effect of coumarin 5 on oxytocin-induced uterine smooth muscle contraction. Thus, coumarin 5 could play a role in relaxing the uterine smooth muscle via the α receptor, rather than the H1 receptor, L-type Ca2+ channel, or prostaglandin release.
Reference
[3] Hajagos-Tóth, J., Bóta, J., Ducza, E., Samavati, R., Borsodi, A., Benyhe, S., Gáspár, R., 2016, The effects of progesterone on the alpha2-adrenergic receptor subtypes in late-pregnant uterine contractions in vitro. Reprod Biol Endocrinol. 14, 33.
[4] Roberts, J.M., Riemer, R.K., Bottari, S.P., Wu, Y.Y., Goldfien, A., 1989, Hormonal regulationmyometrial adrenergic responses: the receptor and beyond, J Dev Physiol. 11, 125–134.
[5] Gáspár, R., Gál, A., Gálik, M., Ducza, E., Minorics, R., Kolarovszki-Sipiczki, Z.., Klukovits, A., Falkay, G., 2007, Different roles of α2-adrenoceptor subtypes in non-pregnant andlate-pregnant uterine contractility in vitro in the rat. Int. 51: 311-318. https://doi.org/10.1016/j.neuint.2007.06.029.
Comment 6: Page 8: what is the meaning of “stabilizing membrane potential? Is really difficult to change membrane potential after the addition of 60 mM KCl.
Response 6: Special thanks for the comment. After the addition of 60 mM KCl, the membrane potential will change. In the manuscript, “the stable membrane potential” refers to the potential recovery. This expression is not accurate and has been revised.
Comment 7: The figure legends are insufficient. Essential details are missing, making it necessary to consult the methodology section for proper interpretation.
Response 7: Thank you for this helpful suggestion. We have revised the figure legends.
Comment 8: Provide a rationale for the selection of VER, DIP, PHE, and IND for the mechanistic studies.
Response 8: Special thanks for your good suggestion. The contraction of uterine smooth muscle is facilitated by pivotal mechanisms involving L-type calcium channels, α-receptors, histamine H1 receptors, and the synthesis and exocytosis of prostaglandins. Voltage-dependent calcium channels, specifically the L-type calcium ion channels, are predominantly found in the smooth muscle cells of uterus, and fluctuations in calcium ion concentration are pivotal in mediating the contractions of these muscle cells. VER acts as a L-type calcium channel blocker, and its application in studies mainly aims to ascertain the mechanism of action [6].
Upon binding of its agonist to the α receptor, this interaction facilitates an increase in intracellular calcium ion levels and the activation of protein kinase through the engagement of associated proteins, thereby driving smooth muscle contraction. The α receptor antagonists such as PHE are chosen to explore the potential role of the α receptor within the drug’s mechanism [7].
After binding to agonists, H1 receptors initiate a signaling cascade that leads to intracellular calcium release. This process is mediated through inositol triphosphate (IP3), which binds to IP3 receptors on the endoplasmic reticulum (ER). The binding of IP3 to these receptors opens calcium channels in the ER membrane, facilitating the release of calcium ions into the cytoplasm. This increase in intracellular calcium ion concentration can trigger a variety of cellular responses, encompassing muscle contraction. Thus, DIP, as an H1 receptor antagonist, was used to investigate whether the drug's mode of action is linked to the H1 receptor [8].
Uterine smooth muscle cells serve as targets for prostaglandin synthetase, facilitating the synthesis of diverse prostaglandin compounds under specific circumstances. This process is instrumental in modulating the contraction and relaxation of uterine smooth muscles. To explore the potential connection between the pharmacological action of a drug and prostaglandins, inhibitors of prostaglandin synthetase, such as IND, were often selected for investigation [9].
Reference
[6] Liu, H., Zhu, T., Ma, Y., Qu, S., 2003, Effect of erythromycin on contractile response of uterine smooth muscle strips in non-pregnant rats, Pol. J. Pharmacol. 55, 57–62.
[7] Xu, J., Wang, W., Qin, X., 2005, Different mechanisms mediated the potentiation uterine smooth muscle caused by Polygala tenuifolia decoction progesterone in virginal rats, Chin. J. Clin. Mater. Med. 30, 295–297.
[8] Shih, H.C., Yang, L.L., 2012, Relaxant effect induced by wogonin from Scutellaria baicalensis on rat isolated uterine smooth muscle, Pharm. Biol. 50, 760–765.
[9] Wong, J., Chiang, Y.F., Shih, Y.H., Chiu, C.H., Chen, H.Y., Shieh, T.M., Wang, K.L., Huang, T.C., Hong, Y.H., Hsia, S.M., 2020, Salvia sclarea L. essential oil extract and its antioxidative phytochemical sclareol inhibit oxytocin-induced uterine hypercontraction dysmenorrhea model by inhibiting the Ca2+-MLCK-MLC20 signaling cascade: An ex vivo and in vivo study, Antioxidants, 9, 991.
Reviewer 2 Report
Comments and Suggestions for Authors
This study shows how different types of coumarins can induce different responses in smooth muscle cells. The outcomes obtained are quite novel. The presentation of results was properly demonstrated and explained, and the discussion and conclusion were fairly clear. However, I have a few comments:
1. The abstract is way too long, not meeting the guidelines. I suggest shortening it.
2. Please be aware that some abbreviations in the manuscript that do not present the full name.
3. Why is only the effect of coumarin 50µM analyzed in section 2.2.2, but in later studies 5µM and 10µM are also analyzed? The same consideration also applies to section 2.3.2.
4. I would make the discussion more complete by putting a more in-depth comparison between Coumarin 1 and 5.
Author Response
Comment 1: The abstract is way too long, not meeting the guidelines. I suggest shortening it.
Response 1: Thank you for this suggestion. We have shortened the abstract.
Comment 2: Please be aware that some abbreviations in the manuscript that do not present the full name.
Response 2: Thank you for your suggestion. We have revised them.
Comment 3: Why is only the effect of coumarin 50µM analyzed in section 2.2.2, but in later studies 5µM and 10µM are also analyzed? The same consideration also applies to section 2.3.2.
Response 3: Thank you for the comment. In section 2.2.1, the experimental results showed that the effect of coumarin 1 at 50 μM was the most significant among the three concentrations (1, 10, and 50 μM). Subsequently, the concentration of 50 μM was chosen, making it easy to explore mechanisms by pretreating with inhibitors. The results in Section 2.2.2 showed that the action of coumarin 1 was related to calcium ion channels. Hence, we focused on the correlation of alterations in calcium ion (Ca2+) concentration and the uterine contraction of coumarin 1. Thus, in the studies on mechanisms, only 50 µM was selected for investigation.
Comment 4: I would make the discussion more complete by putting a more in-depth comparison between Coumarin 1 and 5.
Response 4: Special thanks for the good suggestion. In this study, motherwort coumarins with different substituents showed opposite effects on uterine smooth muscle. Specifically, coumarins 1 and 2 promoted uterine contraction, whereas coumarins 3–5 had inhibitory effects. We have discussed the structure-activity relationships in Section 3 (Discussion). Comparisons of their structures and effects indicated that the contractile activity of furocoumarins in motherwort is closely related to the substitution of OCH3-5. Then, we chosen two coumarins (1 and 5) with the best effects for further studies. The results showed that coumarin 1 could activate the L-type Ca2+ channel and α-receptor, whereas coumarin 5 was an α receptor antagonist. However, it proves challenging to make a more in-depth discussion when the two active compounds have distinct structures and opposing effects and mechanisms. We hope to find more evidence and association between two coumarins 1 and 5 in further studies.
Round 2
Reviewer 1 Report
Comments and Suggestions for Authors
The revised manuscript submitted by Yun-Qiu Fan et al. about the effects of coumarins derived from Leonurus japonicus Houtt. (Chinese motherwort) on uterine smooth muscle (USM) represents a rewritten version, in which figure legends and other minor suggestions were corrected. However, this reviewer believes that there are some major concerns that must be addressed.
1- Thank you for clarifying the meaning of "mean of contractile tension."
The reviewer understands that changes in amplitude must be included in the present manuscript. It's evident from the typical recordings that amplitude and frequency are two independent components of contractile behavior. Furthermore, pharmacological interventions affected one or both of these components in a variable manner. The changes in mean contractile tension can be explained as a combination of the effects on rate and amplitude, making it a valuable indicator of the overall process.
However, it is unclear why the authors switched the parameter used in each experimental series. In Figures 1 and 3, results are presented as mean contractile tension. In Figures 4 to 8, changes in the rate of contractile tension are shown. Figure 2 presents both parameters together. Additionally, in Figure 7, results are displayed as changes in rate, which is impossible to measure after the addition of KCl 60 mM, as this intervention transforms the transient contraction into a tonic contraction.
/span/span/p p class="MsoListParagraphCxSpFirst" style="text-indent: 0pt"span class="fontstyle01"span lang="EN-US" style="mso-fareast-font-family: Arial"span style="mso-list: Ignore"2-span style="font: 7.0pt 'Times New Roman'" /span/span/span/spanAs noted in the initial review, it's inaccurate to conclude a concentration-dependent effect of coumarins solely based on comparisons to the vehicle control. The authors should use ANOVA to compare the effects of different concentrations. Therefore, the following conclusion should be removed:
- Page4: “As shown in Figure 2, coumarin 1 at concentrations of 1, 10, and 50 μM significantly promoted the uterine smooth muscle contraction in a dose-dependent manner.”
- Page 5: First, in a Ca2+-free solution, the excitatory effect of coumarin 1 on rat uterine smooth muscle was enhanced as the concentration increased (Figure 4).”
- Page 9: After treatment with coumarin 5 at 1, 10, and 50 μM, the contractile tension was markedly inhibited in a dose-dependent manner, with inhibition rates of 45.45 ± 3.40% (P < 0.01 vs. control), 56.93 ± 4.10% (P < 0.01 vs. control), and 74.84 ± 7.65% (P< 0.01 vs. control), respectively”
3- The authors should discuss why coumarins 1 and 5 had opposing effects on USM contractions, despite acting on the same receptor type, alpha-adrenergic, in the absence of an endogenous agonist. It is evident that neither compound could be a competitive antagonist, as this mechanism requires the presence of an agonist. Two potential explanations could account for these contrasting results: the existence of alpha receptor subtypes or the intrinsic activity of the receptor targeted by the two coumarins.
4- Figure 7 suggests that coumarin 5 inhibits calcium entry through voltage-operated calcium channels. However, Figure 6 shows that pretreatment with verapamil, a calcium channel blocker, did not affect the inhibition of contraction induced by coumarin 5. Please discuss this contradictory result.
Author Response
Comment 1: Thank you for clarifying the meaning of "mean of contractile tension." The reviewer understands that changes in amplitude must be included in the present manuscript. It's evident from the typical recordings that amplitude and frequency are two independent components of contractile behavior. Furthermore, pharmacological interventions affected one or both of these components in a variable manner. The changes in mean contractile tension can be explained as a combination of the effects on rate and amplitude, making it a valuable indicator of the overall process. However, it is unclear why the authors switched the parameter used in each experimental series. In Figures 1 and 3, results are presented as mean contractile tension. In Figures 4 to 8, changes in the rate of contractile tension are shown. Figure 2 presents both parameters together.
Additionally, in Figure 7, results are displayed as changes in rate, which is impossible to measure after the addition of KCl 60 mM, as this intervention transforms the transient contraction into a tonic contraction.
Response: Thank you for these good comments. In all Figures, the inhibition rates of contractile tension were used to present the results in each experimental series. Meanwhile, we would like to explain why some experimental results did not show contractile activity. Both inhibition rates of contractile tension and activity are important parameters for evaluating the efficacy of contractile behavior. However, we found that the erratic fluctuations in contractile activity were observed in certain experiments, especially in the ones pretreated with inhibitors. Generally, the irregularity in contractile activity is attributed to the variations in experimental conditions influencing the sensitivity of the muscle strips to the substances used. When the uterine muscle strips are concurrently exposed to multiple compounds, potential interactions among these substances can lead to unpredictable contractile activities. Therefore, the calculated inhibition rates of contractile activity exhibited irregularities, rendering it inappropriate and meaningless to compare the data.
KCl is employed to induce a high extracellular potassium environment, thereby facilitating the depolarization of the membrane potential. This method is commonly used to investigate whether a substance exerts its effects through the modulation of voltage-dependent calcium channels. As shown in Figure 7, after the addition of KCl (60 mM), the frequency could not be determined, because this intervention transformed the transient contraction into a tonic contraction. Thus, the inhibition rate of contractile activity couldn’t be calculated [inhibition rate of contractile activity = (model mean of contractile tension × frequency) – (test mean of contractile tension × frequency)/ (model mean of contractile tension × frequency) × 100%]. In these experiments, only contractile tension recorded by the system can be used to reflect the contractile behavior (Antioxidants, 2020, 9: 991; J. Ethnopharmacol. 2021, 269: 113713).
Comment 2: As noted in the initial review, it's inaccurate to conclude a concentration-dependent effect of coumarins solely based on comparisons to the vehicle control. The authors should use ANOVA to compare the effects of different concentrations. Therefore, the following conclusion should be removed:
-Page4: “As shown in Figure 2, coumarin 1 at concentrations of 1, 10, and 50 μM significantly promoted the uterine smooth muscle contraction in a dose-dependent manner.”
-Page 5: First, in a Ca2+-free solution, the excitatory effect of coumarin 1 on rat uterine smooth muscle was enhanced as the concentration increased (Figure 4).”
-Page 9: After treatment with coumarin 5 at 1, 10, and 50 μM, the contractile tension was markedly inhibited in a dose-dependent manner, with inhibition rates of 45.45 ± 3.40% (P < 0.01 vs. control), 56.93 ± 4.10% (P < 0.01 vs. control), and 74.84 ± 7.65% (P< 0.01 vs. control), respectively”
Response: Thank you for this suggestion. We have revised these descriptions.
Comment 3: The authors should discuss why coumarins 1 and 5 had opposing effects on USM contractions, despite acting on the same receptor type, alpha-adrenergic, in the absence of an endogenous agonist. It is evident that neither compound could be a competitive antagonist, as this mechanism requires the presence of an agonist. Two potential explanations could account for these contrasting results: the existence of alpha receptor subtypes or the intrinsic activity of the receptor targeted by the two coumarins.
Response: Special thanks for the helpful comment. We have added the discussion in the manuscript according to the good suggestions. In this study, it has been elucidated that the opposite activities of the coumarins 1 and 5 are both associated with the α receptor. It is known that two primary subtypes of α-receptor, α1 and α2, play distinct roles in uterine smooth muscle. Activation of the α1 receptor often triggers muscle contractions, while the α2 receptor may prevent contractions or modulate their response, exhibiting a more complex regulatory function. Therefore, it is plausible that the mechanisms of coumarins 1 and 5 might be attributed to their interaction with distinct subtypes of α receptors or the intrinsic activity of the receptors targeted by the two coumarins.
Comment 4: Figure 7 suggests that coumarin 5 inhibits calcium entry through voltage-operated calcium channels. However, Figure 6 shows that pretreatment with verapamil, a calcium channel blocker, did not affect the inhibition of contraction induced by coumarin 5. Please discuss this contradictory result.
Response:Thank you very much for the good comment. We are sorry for our inaccurate statements. In section 2.3.3.1, “L-type Ca2+ channels” has been revised as “voltage-gated calcium channels”.
Firstly, KCl was used to create a high K+ condition for depolarization of the membrane potential, which could open the voltage-dependent Ca2+ channel, causing extracellular Ca2+ influx. As shown in Figure 7, coumarin 5 inhibits the KCl-induced contraction of uterine smooth muscle. This result suggests that coumarin 5 could affect the voltage-gated calcium channels to inhibit extracellular Ca2+ influx. Verapamil is a blocker of L-type calcium channels which belong to voltage-gated calcium channels. As shown in Figure 6, the results indicated that the inhibitory effect of coumarin 5 on uterus contraction didn’t involve the L-type calcium channels. Taken together, coumarin 5 is capable of triggering the voltage-gated calcium channels, but excluding the L-type calcium channels.
